# AGLP: A Graph Learning Perspective for Semi-supervised Domain Adaptation

## Abstract

In semi-supervised domain adaptation (SSDA), the model aims to leverage partially labeled target domain data along with a large amount of labeled source domain data to enhance its generalization capability for the target domain. A key advantage of SSDA is its ability to significantly reduce reliance on labeled data, thereby lowering the costs and time associated with data preparation. Most existing SSDA methods utilize information from domain labels and class labels but overlook the structural information of the data. To address this issue, this paper proposes a graph learning perspective (AGLP) for semi-supervised domain adaptation. We apply the graph convolutional network to the instance graph which allows structural information to propagate along the weighted graph edges. The proposed AGLP model has several advantages. First, to the best of our knowledge, this is the first work to model structural information in SSDA. Second, the proposed model can effectively learn domain-invariant and semantic representations, reducing domain discrepancies in SSDA. Extensive experimental results on multiple standard benchmarks demonstrate that the proposed AGLP algorithm outperforms state-of-the-art semi-supervised domain adaptation methods.

## 1 Introduction

Domain Adaptation (DA) Venkateswara et al. (2017); Peng et al. (2019); Berthelot et al. (2021) is a critical machine learning approach aimed at addressing the issue of training and test data originating from two related but distinct domains. These domains are typically referred to as the source domain and the target domain. In many practical applications, the source domain contains a wealth of labeled data, while the target domain may have only a few labels or even none at all. This discrepancy often leads to a significant drop in model performance when directly transferring a model from the source domain to the target domain. Much of the research has focused on Unsupervised Domain Adaptation (UDA). In UDA scenarios, researchers cannot access labels from the target domain, requiring models to rely on knowledge from the source domain and unlabeled data from the target domain for learning. In recent years, Semi-Supervised Domain Adaptation (SSDA) has emerged as a focal point of research. Unlike UDA, SSDA Jiang et al. (2020); Singh (2021); Berthelot et al. (2021) allows researchers to access a small number of labeled samples in the target domain, providing the model with richer learning information. By combining the abundant labeled data from the source domain with the limited labeled data from the target domain, SSDA can more effectively capture the underlying structural relationships between the domains, thereby improving the model's performance and adaptability.

Prior SSDA methods can be broadly categorized into three groups: 1) statistical discrepancy minimization methods Berthelot et al. (2021); Li & Zhang (2018), which utilize statistical regularizations to explicitly reduce the cross-domain distribution discrepancy; 2) adversarial learning methods Jiang et al. (2020); Singh (2021), which aim to learn domain-invariant representations across two domains using adversarial techniques; and 3) multi-task learning methods Li et al. (2019); Qi et al. (2024), which focus on simultaneously learning multiple related tasks to share knowledge and improve the model's generalization ability.

Indeed, these SSDA methods have achieved some success, but the main technical challenge in SSDA lies in how to formally reduce the distribution discrepancy between different domains, typically the labeled source domain and the sparsely labeled target domain. There is little literature addressing

the significant enhancement of the adaptation capability of source-supervised classifiers, which is crucial for SSDA problems, as shown in Figure 2. To achieve classifier adaptation, He et al.He et al. (2020) propose a novel classification-aware semi-supervised translator that effectively addresses the large gap between heterogeneous domains at the pixel level. Saito et al.Saito et al. (2019a) tackle the SSDA setting by proposing a novel Minimax Entropy approach that adversarially optimizes an adaptive few-shot model. The domain classifier is trained to determine whether a sample comes from the source domain or the target domain. The feature extractor is trained to minimize classification loss while maximizing domain confusion loss. Through the principled lens of adversarial training, it appears possible to obtain domain-invariant yet discriminative features. All of these methods overlook the aspect of learning domain-invariant features from the perspective of data structure.

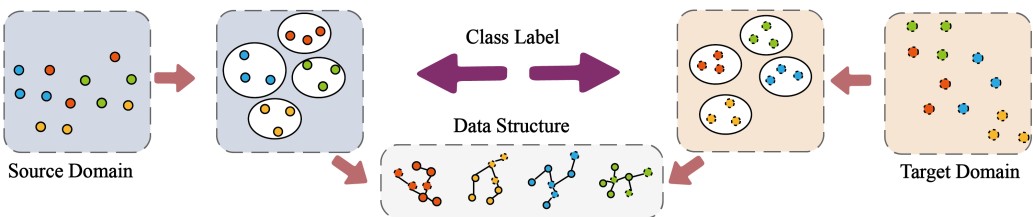

Figure 1: Illustration of our AGLP. The data structure is constructed to build graph information.

To address the above issues, we propose an end-to-end Graph Convolutional Adversarial Network (GCAN) aimed at achieving semi-supervised domain adaptation. This network enhances adaptability by jointly modeling data structure and domain labels within a unified deep model. Inspired by graph neural networks, we construct a densely connected instance graph using the CNN features of samples, based on the similarity of their structural characteristics. Each node corresponds to the CNN features of a sample extracted by a standard convolutional network. Next, we apply a Graph Convolutional Network (GCN) to the instance graph, allowing structural information to propagate along the weighted graph edges that can be learned from the designed network. During the class centroid alignment process, we constrain the centroids of different classes to gradually move closer as iterations increase, enabling the learned representations to effectively encode class label information. This results in tighter embeddings for samples with the same category label in the feature space. Our model introduces a class alignment loss to achieve this goal and employs a moving centroid strategy to mitigate the influence of incorrect pseudo-labels. By modeling this alignment mechanism, the deep network can generate domain-invariant and highly discriminative semantic representations. The main contributions of this work can be summarized as follows.

- We propose a graph learning perspective (AGLP) by modeling data structure and domain label for semi-supervised domain adaptation. To the best of our knowledge, this is the first work to model graph information for semi-supervised domain adaptation.

- The proposed alignment mechanisms can learn domain-invariant and semantic representations effectively to reduce the domain discrepancy for SSDA.

- Extensive experimental results on several standard benchmarks demonstrate that the proposed AGLP algorithm performs favorably against state-of-the-art SSDA methods.

## 2 METHODS

### 2.1 PRELIMINARIES

#### 2.1.1 SEMI-SUPERVISED DOMAIN ADAPTATION

**Semi-Supervised Domain Adaptation (SSDA)** aims to learn a classifier for the target domain, given labeled data $S = \{(x_i^s, y_i^s)\}_{i=1}^{N_s}$ from a source domain, along with both unlabeled data $U = \{(x_i^u)\}_{i=1}^{N_u}$ and labeled data $L = \{(x_i^l, y_i^l)\}_{i=1}^{N_l}$ from the target domain Saito et al. (2019a); Li & Hospedales (2020); Berthelot et al. (2021); Wang et al. (2023). The primary goal of SSDA is to leverage these data subsets to train a feature extractor $\mathcal{F}(\cdot)$ and a classifier $\mathcal{C}(\cdot)$, facilitating the migration of learned knowledge from the source domain to the target domain, while minimizing the

risk of migration loss. SSDA can be viewed as a more flexible yet practical extension of Unsupervised Domain Adaptation (UDA)Yue et al. (2023); Litrico et al. (2023), where some labeled data from the target domain is available. Typically, SSDA algorithms utilize a combination of three loss functions:

$$\mathcal{L}_{\text{SSDA}} = \mathcal{L}_s + \mathcal{L}_\ell + \mathcal{L}_u \tag{1}$$

where $\mathcal{L}_s$ represents the loss from the source data, $\mathcal{L}_\ell$ and $\mathcal{L}_u$ correspond to the losses from the labeled and unlabeled target data, respectively.

To train the model effectively using supervision from both the source and target domains, most existing SSDA methods Yu & Lin (2023); Li & Hospedales (2020); Berthelot et al. (2021) include the following standard cross-entropy loss for all labeled data:

$$\mathcal{L}_\ell = \mathcal{L}_{CE} = - \sum_{(x,y)\in\mathcal{S}\cup\mathcal{L}} y \log(p(x)) \tag{2}$$

In Eq. 2, $(x, y)$ represents the data points and their corresponding labels from the source domain $\mathcal{S}$ and the labeled target domain $\mathcal{L}$. The cross-entropy loss encourages the model to minimize the negative log-likelihood of the predicted probability $p(x)$ with respect to the true label $y$, thereby facilitating effective learning from both domains.

### 2.1.2 CROSS-DOMAIN ADAPTIVE CLUSTERING (CDAC)

Inspired by a recent well-known method CDAC Li et al. (2021), we consider improving model performance from the perspective of cross-domain clustering. CDAC introduce an adversarial adaptive clustering loss in SSDA to align target domain features by forming clusters and aligning them with source domain clusters. This loss computes pairwise feature similarities among target samples and ensures that samples with similar features share the same predicted class labels. Pairwise similarities are used to define binary pseudo-labels for sample pairs, $s_{ij} = 1$ for similar pairs and $s_{ij} = 0$ otherwise, based on the top-$k$ ranked feature elements:

$$s_{ij} = \mathbf{1}\{\text{topk}(G(x_i^u)) = \text{topk}(G(x_j^u))\} \tag{3}$$

where $\text{topk}(\cdot)$ denotes the top-$k$ indices of rank ordered feature elements and we set $k = 5$. And $\mathbf{1}\{\cdot\}$ is an indicator function.

The adversarial adaptive clustering loss $\mathcal{L}_{AAC}$ is formulated as:

$$\mathcal{L}_{AAC} = - \sum_{i=1}^{M} \sum_{j=1}^{M} s_{ij} \log(P_i^T P_j) + (1 - s_{ij}) \log(1 - P_i^T P_j) \tag{4}$$

where $M$ is the number of unlabeled target samples in each mini-batch and $P_i = p(x_i^u) = \sigma(F(G(x_i^u)))$ represents the prediction of an image $x_i^u$ in the mini-batch. Also, $P_i' = p(x_i') = \sigma(F(G(x_i')))$ indicates the prediction of a transformed image $x_i'$, which is an augmented version of $x_i^u$ using a data augmentation technique. The inner product $P_i^T P_i'$ in Eq. 4 is used as a similarity score, which predicts whether image $x_i^u$ and the transformed version of image $x_i'$ share the same class label or not.

To address the lack of labeled target samples, CDAC apply pseudo labeling, retaining high-confidence pseudo-labels to increase the number of labeled target samples. Pseudo labels are generated by feeding an unlabeled image $x_j^u$ into the model, with the prediction $P_j = p(x_j^u)$ converted into a hard label $\tilde{y}_j^u = \arg\max(P_j)$. The final loss for pseudo-labeling is defined as:

$$\mathcal{L}_{PL} = - \sum_{j=1}^{M} \mathbf{1}\{\max(P_j) \geq \tau\} \tilde{y}_j^u \log(p_j'') \tag{5}$$

where $P_j'' = p(x_j'') = \sigma(F(G(x_j'')))$ denotes the model prediction of the transformed image $x_j''$, and $\tau$ is a scalar confidence threshold that determines the subset of pseudo labels that should be retained for model training.

To improve the input diversity of our model, we create two different transformed versions of each unlabeled image in the target domain to implement the adversarial adaptive clustering loss and the pseudo-labeling loss, respectively. Therefore, CDAC employ a consistency loss, $\mathcal{L}_{Con}$, to keep the model predictions on these two transformed images consistent:

$$\mathcal{L}_{Con} = w(t) \sum_{j=1}^{M} \|P_j' - P_j''\|^2 \tag{6}$$

$w(t) = \nu e^{-5(1-\frac{t}{T})^2}$ is a ramp-up function used in with the scalar coefficient $\nu$, the current time step $t$, and the total number of steps $T$ in the ramp-up process. So, the $\mathcal{L}_u$ is:

$$\mathcal{L}_u = \mathcal{L}_{Con} + \mathcal{L}_{PL} + \mathcal{L}_{AAC} \tag{7}$$

### 2.1.3 SOURCE LABEL ADAPTATION (SLA)

In SSDA, accessing only a few labeled target instances can lead to overfitting. To mitigate this, SLAYu & Lin (2023) employs a prototypical network (ProtoNet) to address the few-shot problem. Given a dataset, $\{(x_i, y_i)\}_{i=1}^{N}$ and a feature extractor $f$, the prototype of class $k$ is defined as the mean of the feature representations for all samples belonging to class $k$:

$$c_k = \frac{1}{N_k} \sum_{i=1}^{N} \mathbf{1}\{y_i = k\} \cdot f(x_i). \tag{8}$$

The set of all class prototypes is denoted as $C_f = \{c_1, \ldots, c_K\}$. A ProtoNet is defined using these class prototypes as:

$$P_{C_f}(x_i)_k = \frac{\exp(-d(f(x_i), c_k) \cdot T)}{\sum_{j=1}^{K} \exp(-d(f(x_i), c_j) \cdot T)} \tag{9}$$

where $d(\cdot)$ is a distance function in the feature space, typically Euclidean distance, and $T$ controls the smoothness of the output distribution.

To adapt to the target domain, labeled target centers $C_f'$ are computed from labeled target data. The ProtoNet with labeled target centers $P_{C_f'}$ serves as a label adaptation model. However, since the number of labeled target samples is limited, the ideal centers $C_f^*$ should be estimated from both labeled and pseudo-labeled data. Pseudo centers $\tilde{C}_f$ are computed using pseudo-labels for the unlabeled target data, which are predicted as:

$$\tilde{y}_i^u = \arg\max_k g(x_i^u)_k \tag{10}$$

After deriving unlabeled target data with pseudo labels $\{(x_i^u, \tilde{y}_i^u)\}_{i=1}^{|U|}$, we can get pseudo centers $C_f'$ by Eq. 8, and further define a ProtoNet with Pseudo Centers (PPC) $P_{C_f'}$ by Eq. 9.

The ProtoNet with pseudo centers (PPC) $\tilde{P}_{C_f}$ better approximates the ideal centers. Then the updated source label is computed as:

$$y_i^s = (1 - \alpha) \cdot y_i^s + \alpha \cdot P_{\tilde{C}_f}(x_i^s) \tag{11}$$

The source label adaptation loss $\tilde{L}_s$ replaces the standard cross-entropy loss for the source data:

$$\mathcal{L}_s = \tilde{\mathcal{L}}_s(g|S) = \frac{1}{|S|} \sum_{i=1}^{|S|} H(g(x_i^s), \tilde{y}_i^s) \tag{12}$$

The final loss function for SSDA with CDAC SLA is:

$$\mathcal{L}_{CDACSLA} = \tilde{\mathcal{L}}_s(g|S) + \mathcal{L}_{CE} + \mathcal{L}_{AAC} + \mathcal{L}_{PL} + \mathcal{L}_{Con} \tag{13}$$

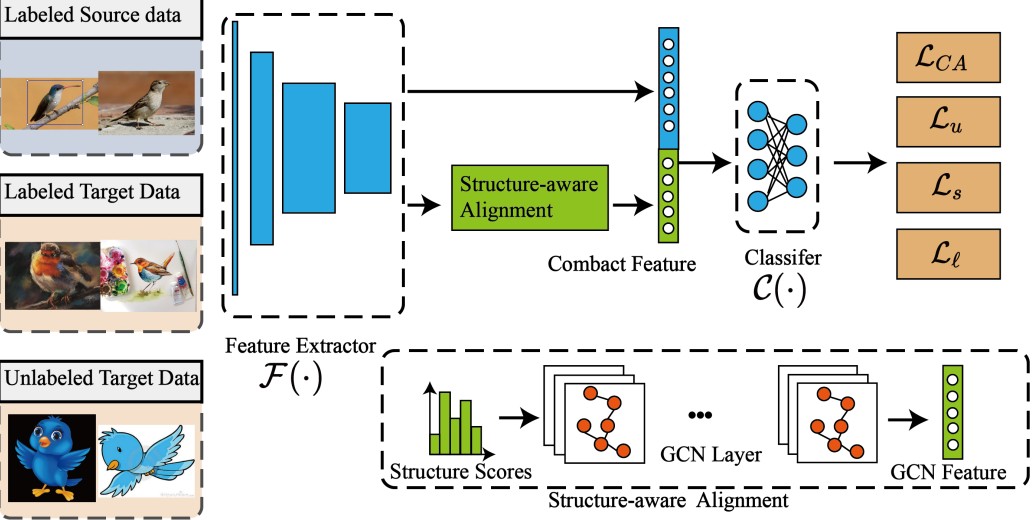

Figure 2: Overall framework of our model.

## 2.2 STRUCTURE-AWARE ALIGNMENT

In traditional domain alignment mechanisms Sun et al. (2023); Li et al. (2023), only global domain statistics are aligned, overlooking the inherent structural information in mini-batch samples. Previous research has focused primarily on modeling data structure in unsupervised domain adaptation (UDA) and has achieved promising results Oza et al. (2023). However, in the context of SSDA, there has been no solution addressing the structural information within mini-batch samples, despite its importance being demonstrated in UDA. To overcome this limitation in SSDA, we propose a structure-aware alignment mechanism that more effectively captures the structural relationships between mini-batch source and target samples.

Our approach begins by utilizing a Data Structure Analyzer (DSA) network to generate structural scores for mini-batch samples. These scores, together with the learned CNN features of the samples, are used to construct a densely connected instance graph. This instance graph is then processed using a Graph Convolutional Network (GCN) Kipf & Welling (2016), which learns features that encode the structural information present in the data.

GCNs are designed to perform hierarchical propagation operations on graphs. Given an undirected graph with $m$ nodes and a set of edges represented by an adjacency matrix $A \in \mathbb{R}^{k \times m}$, the graph convolution's linear transformation is expressed as a graph signal $G \in \mathbb{R}^{k \times m}$, where $G_i \in \mathbb{R}$ represents the feature of the $i$-th node. This is combined with a filter $W \in \mathbb{R}^{k \times c}$ for feature extraction.

$$\mathbf{Z} = \hat{\mathbf{D}}^{-\frac{1}{2}} \hat{\mathbf{A}} \hat{\mathbf{D}}^{-\frac{1}{2}} \mathbf{G}^T \mathbf{W} \tag{14}$$

In our method, the GCN is constructed by stacking multiple graph convolutional layers, each followed by a non-linear activation (e.g., ReLU). Given the adjacency matrix $\hat{A} = A + I$, where $I$ is the identity matrix and $D_{ii} = \sum_j \hat{A}_{ij}$, the output of the GCN is a $c \times m$ matrix $Z$.

To build densely-connected instance graphs for GCN, the graph signal $X$ is generated using a standard convolutional network:

$$G = \mathcal{F}(x_{\text{batch}}) \tag{15}$$

where $x_{\text{batch}}$ represents mini-batch samples. The adjacency matrix $\hat{A}$ is constructed using structure scores $G_{sc}$ produced by a Data Structure Analyzer (DSA) network:

$$\hat{A} = G_{sc}G_{sc}^T, \tag{16}$$

where $G_{sc} \in \mathbb{R}^{w \times h}$, $w$ is the batch size, and $h$ is the dimension of the structure scores.

## 2.3 CLASS CENTROID ALIGNMENT

Domain invariance and structure consistency do not necessarily guarantee discriminability. For example, features of the target class "laptops" may be mapped near features of the source class "screens" while still satisfying domain invariance. To address this, we draw inspiration from UDA Ma et al. (2019), where class label information ensures that features of the same class from different domains are mapped nearby. This motivates our use of class centroid alignment in UDA, following the approach in Ma et al. (2019).

To implement the class centroid alignment, pseudo labels are first assigned using a target classifier $F$, after which centroids are computed for both labeled and pseudo-labeled samples. The centroid alignment objective is defined as:

$$\mathcal{L}_{CA}(\mathcal{X}_S, \mathcal{Y}_S, \mathcal{X}_T, \mathcal{Y}_T) = \sum_{k=1}^{K} \phi(C_S^k, C_T^k), \tag{17}$$

where $C_S^k$ and $C_T^k$ are the centroids of class $k$ in the source and target domains, respectively. The distance measure $\phi(\cdot, \cdot)$ is defined as the squared Euclidean distance $\phi(x, x') = \|x - x'\|^2$. By minimizing the distance between centroids across domains, we ensure that features of the same class are mapped nearby.

## 2.4 IMPLEMENTATION DETAILS

The overall framework of our final model is illustrated in Figure 2. After extracting features from the input, we compute the structural score using Structure-aware Alignment and extract structural features through Graph Convolutional Networks (GCN). These features are then concatenated with the original features to create the final feature representation. Finally, we utilize the final loss for convergence, which is defined as follows:

$$\begin{aligned}
\mathcal{L}_{AGLP} &= \mathcal{L}_{CDACSLA} + \beta\mathcal{L}_{CA}(\mathcal{X}_S, \mathcal{Y}_S, \mathcal{X}_T, \mathcal{Y}_T) \\
&= \tilde{\mathcal{L}}_s(g|S) + \mathcal{L}_{CE} + \mathcal{L}_{AAC} + \mathcal{L}_{PL} + \mathcal{L}_{Con} + \beta\mathcal{L}_{CA}(\mathcal{X}_S, \mathcal{Y}_S, \mathcal{X}_T, \mathcal{Y}_T)
\end{aligned} \tag{18}$$

where $\beta$ is a hyperparameter, which is typically set to 1 in our experiments.

# 3 EXPERIMENTS

## 3.1 EXPERIMENT DATASETS

We evaluate our proposed AGLP framework on two SSDA benchmarks: Office-Home Venkateswara et al. (2017) and DomainNet Peng et al. (2019).

**Office-Home Venkateswara et al. (2017)** is an object recognition benchmark consisting of 15,500 images from 65 classes across four domains: Art (A), Clipart (C), Product (P), and Real World (R). The domain shift primarily results from variations in image styles and perspectives.

**DomainNet Peng et al. (2019)** is a dataset featuring common objects across six different domains, including 345 classes such as bracelets, airplanes, birds, and cellos. The domains include Clipart,

---

**Algorithm 1: AGLP algorithm.**

---

**Input:**

1) Source domain data $S = \{(x_i^s, y_i^s)\}_{i=1}^{N_s}$

2) Unlabeled target data $U = \{(x_i^u)\}_{i=1}^{N_u}$

3) Labeled target data $L = \{(x_i^l, y_i^l)\}_{i=1}^{N_l}$

4) Feature extractor $\mathcal{F}(\cdot)$

5) Classifier $\mathcal{C}(\cdot)$

6) GCN network

1 Initialize all parameters

3 **for** $l \leftarrow 0$ to $L$ **do**

5     Randomly sample a batch of data from $\mathcal{S}, \mathcal{U}, \mathcal{L}$.

7     Use $\mathcal{F}(\cdot)$ to extract features and obtain $G$ as shown in Eq. 15.

9     Obtain the structural information feature $\hat{A}$ by passing $G$ through the DSA in Eq. 16.

11     Concatenate $\hat{A}$ with $G$ and feed the combined features into $\mathcal{C}(\cdot)$.

13     Train $\mathcal{C}(\cdot)$ and $\mathcal{F}(\cdot)$ using the losses $\tilde{\mathcal{L}}_s(g|S), \mathcal{L}_{CE}, \mathcal{L}_{AAC}, \mathcal{L}_{PL}, \mathcal{L}_{Con}$, and $\mathcal{L}_{CA}$.

14 **end**

15 **return** $\mathcal{C}(\cdot), \mathcal{F}(\cdot)$

---

which contains clipart images; Real, comprising photographs and real-world images; Sketch, featuring sketches of tangible objects; Infograph, containing infographics with specific objects; Painting, showcasing artistic representations; and Quickdraw, which consists of drawings made by players worldwide. In line with prior works Yang et al. (2021); Li et al. (2021); Yan et al. (2022), we select four domains—Clipart (C), Painting (P), Real (R), and Sketch (S)—to conduct experiments on 126 classes. For dataset processing, we employ the same sampling strategy for the training and validation sets as utilized in recent studies Yang et al. (2021); Li et al. (2021); Yan et al. (2022). Each dataset is evaluated through both one-pass and three-pass experiments.

### 3.2 COMPARISON METHODS AND SETTINGS

We compare our results with several baselines, including S+T, DANN Ganin et al. (2016), ENT Grandvalet & Bengio (2004), APE Kim & Kim (2020), DECOTA Yang et al. (2021), MME Saito et al. (2019a), MME SLA Yu & Lin (2023), CDAC Li et al. (2021), and CDAC SLA Yu & Lin (2023). Among these, S+T serves as the baseline method for SSDA, where training involves only source data and labeled target data. DANN is a classical unsupervised domain adaptation method, replicated here by training on additional labeled target data. ENT is the standard entropy minimization method originally designed for semi-supervised learning.

Our framework can be applied to various state-of-the-art methods. To verify the effectiveness of our approach, we select CDAC SLA Yu & Lin (2023) as the baseline. For a fair comparison, we adopt ResNet34 He et al. (2016) as the backbone network. The backbone network is pre-trained on the ImageNet1K dataset Deng et al. (2009), and we follow the same model architecture, batch size, learning rate scheduler, optimizer, weight decay, and initialization strategies as in previous works Li et al. (2021); Saito et al. (2019a). For MME and CDAC, we use the same hyperparameters as recommended in their original papers. For SLA, we set the mixing ratio $\alpha$ to 0.3 and the temperature parameter $T$ to 0.6. The update interval is set to 500. For MME, the warmup parameter $W$ is 500 on Office-Home and 3000 on DomainNet, while for CDAC, $W$ is 2000 on Office-Home and 5000 on DomainNet. After the warmup phase, we reset the learning rate scheduler to allow label adaptation loss updates at a higher learning rate. All hyper-parameters are fine-tuned through a validation process. For each sub-task, we conduct three experiments. The hyper-parameters for the other comparative models are kept identical to those in their original papers.

The parameters we use in the structure alignment module are as follows: the input channels for the GCN are set to 1000, with hidden channels set to 256 and dropout set to 0.2. The output channels are configured to 200 for the Office-Home dataset and 25 for DomainNet. The number of GCN layers is set to 4 for the Office-Home dataset and 8 for DomainNet. Additionally, the hyper-parameter $\beta$ in

the class centroid alignment section is uniformly set to 1 throughout the paper. A robustness analysis of these parameters is provided in the supplementary materials.

Table 1: In the 3-Shot comparison experiments conducted on the Office-Home dataset, the best results are highlighted in bold.

| Method | A→C | A→P | A→R | C→A | C→P | C→R | P→A | P→C | P→R | R→A | R→C | R→P | Avg |
|---|---|---|---|---|---|---|---|---|---|---|---|---|---|
| S+T | 54.0 | 73.1 | 74.2 | 57.6 | 72.3 | 68.3 | 63.5 | 53.8 | 73.1 | 67.8 | 55.7 | 80.8 | 66.2 |
| DANNGanin et al. (2016) | 54.7 | 68.3 | 73.8 | 55.1 | 67.5 | 67.1 | 56.6 | 51.8 | 69.2 | 65.2 | 57.3 | 75.5 | 63.5 |
| ENTGrandvalet & Bengio (2004) | 61.3 | 79.5 | 79.1 | 64.7 | 79.1 | 70.2 | 62.6 | 85.7 | 71.9 | 73.4 | 66.4 | 86.2 | 74.0 |
| APEKim & Kim (2020) | 63.9 | 81.1 | 80.2 | 66.6 | 79.9 | 76.8 | 67.1 | 65.2 | 82.0 | 74.0 | 70.4 | **87.7** | 75.7 |
| DECOTAYang et al. (2021) | 64.0 | 81.8 | 80.5 | 68.0 | 83.2 | 79.0 | 69.9 | 68.0 | 82.1 | 74.0 | 70.4 | **87.7** | 75.7 |
| MMESaito et al. (2019a) | 63.6 | 79.0 | 79.7 | 67.2 | 79.6 | 76.6 | 65.5 | 64.6 | 80.1 | 71.3 | 64.6 | 85.5 | 73.1 |
| MME SLAYu & Lin (2023) | 65.9 | 81.1 | 80.5 | 69.2 | 81.9 | 79.4 | 69.7 | 67.4 | 81.9 | 74.7 | 68.4 | 87.4 | 75.6 |
| CDACLi et al. (2021) | 66.7 | 79.0 | 83.6 | 66.7 | 78.0 | 80.0 | 64.1 | 67.2 | 86.2 | 68.7 | 69.7 | 86.2 | 74.7 |
| CDAC SLAYu & Lin (2023) | 65.6 | 81.4 | 81.1 | 68.2 | **82.1** | 80.1 | 67.7 | 68.9 | 82.6 | 69.0 | 69.7 | 86.3 | 75.2 |
| AGLP(Ours) | **68.9** | **85.1** | **87.2** | **70.3** | **82.1** | **81.0** | **70.3** | **71.3** | **88.2** | **71.3** | **70.3** | 85.6 | **77.6** |

Table 2: In the 1-Shot comparison experiments conducted on the Office-Home dataset, the best results are highlighted in bold.

| Method | A→C | A→P | A→R | C→A | C→P | C→R | P→A | P→C | P→R | R→A | R→C | R→P | Avg. |
|---|---|---|---|---|---|---|---|---|---|---|---|---|---|
| S+T | 50.9 | 69.8 | 73.8 | 56.3 | 68.1 | 70.0 | 57.2 | 48.3 | 74.4 | 66.2 | 52.1 | 78.6 | 63.8 |
| DANNGanin et al. (2016) | 52.3 | 67.9 | 73.9 | 54.1 | 66.8 | 69.2 | 55.7 | 51.9 | 68.4 | 64.5 | 53.1 | 74.8 | 62.7 |
| ENTGrandvalet & Bengio (2004) | 52.9 | 75.0 | 76.7 | 63.2 | 73.6 | 70.4 | 53.6 | 81.9 | 67.9 | 72.5 | 60.7 | 81.6 | 68.9 |
| APEKim & Kim (2020) | 53.9 | 76.1 | 75.2 | 63.6 | 69.8 | 72.3 | 58.3 | 78.6 | 72.5 | 71.3 | 56.0 | 79.4 | 64.8 |
| DECOTAYang et al. (2021) | 42.1 | 68.5 | 72.6 | 60.3 | 70.4 | 71.3 | 48.8 | 76.9 | 71.2 | 70.7 | 60.0 | 79.4 | 64.8 |
| MMESaito et al. (2019a) | 59.6 | 75.5 | 77.8 | 65.7 | 74.5 | 74.8 | 64.7 | 57.4 | 79.2 | 71.2 | 61.9 | 82.8 | 70.4 |
| MME SLAYu & Lin (2023) | 62.1 | 76.3 | 78.6 | **67.5** | 77.1 | 75.1 | 66.7 | 59.9 | 80.0 | **72.9** | 64.1 | 83.8 | 72.0 |
| CDACLi et al. (2021) | 61.2 | 75.9 | 78.5 | 64.5 | 75.1 | 75.3 | 64.6 | 59.3 | 80.0 | 72.7 | 61.9 | 83.1 | 71.0 |
| CDAC SLAYu & Lin (2023) | 61.4 | 77.8 | 79.2 | 66.9 | **76.2** | 75.9 | 66.3 | 60.6 | 80.5 | 71.6 | 65.6 | 84.3 | 72.2 |
| AGLP(Ours) | **66.2** | **84.1** | **85.6** | 67.2 | 75.5 | **76.8** | 68.2 | 62.1 | 84.6 | 71.9 | **69.7** | 84.6 | 74.7 |

Table 3: In the 3-Shot comparison experiments conducted on the DomainNet dataset, the best results are highlighted in bold.

| Method | R→C | R→P | P→C | C→S | S→P | R→S | P→R | Avg. |
|---|---|---|---|---|---|---|---|---|
| S+T | 60.0 | 62.2 | 59.4 | 55.0 | 59.5 | 50.1 | 73.9 | 60.0 |
| DANNGanin et al. (2016) | 59.8 | 62.8 | 59.6 | 55.4 | 59.9 | 54.9 | 72.2 | 60.7 |
| ENTGrandvalet & Bengio (2004) | 71.0 | 69.2 | 71.1 | 60.0 | 62.1 | 61.1 | 78.6 | 67.6 |
| APEKim & Kim (2020) | 76.6 | 72.1 | 76.7 | 63.1 | 66.1 | 67.8 | 79.4 | 71.7 |
| DECOTAYang et al. (2021) | 80.4 | 75.2 | 78.7 | 68.6 | 72.7 | 71.9 | 81.5 | 75.6 |
| MMESaito et al. (2019a) | 72.2 | 69.7 | 71.7 | 61.8 | 66.8 | 61.9 | 78.5 | 68.9 |
| MME SLAYu & Lin (2023) | 73.3 | 70.1 | 72.7 | 63.4 | 67.3 | 63.9 | 79.6 | 70.0 |
| CDACLi et al. (2021) | 79.6 | 75.1 | 79.3 | 69.9 | 73.4 | 72.5 | 81.9 | 76.0 |
| CDAC SLAYu & Lin (2023) | 80.9 | 75.2 | 80.2 | 70.8 | 72.4 | **73.5** | 82.5 | 76.5 |
| AGLP(Ours) | **82.0** | **76.4** | **81.4** | **71.6** | **73.4** | **73.5** | **82.6** | **77.3** |

## 3.3 COMPARATIVE EXPERIMENTS

**Comparative Experiments on Office-Home:** We conducted 1-Shot and 3-Shot experiments on the Office-Home dataset, with results summarized in Table. 1 and Table. 2. In the Office-Home 3-Shot experiment, our method, AGLP, demonstrated excellent performance across multiple transfer tasks, achieving an average accuracy of 77.6%, surpassing all other methods, including the baseline CDAC SLA. In the more stringent Office-Home 1-Shot setting, AGLP maintained its lead with an average accuracy of 74.7%, showcasing robust performance even under data-scarce conditions. The AGLP method exhibited significant improvements in both the 3-Shot and 1-Shot experiments, exceeding the previous state-of-the-art baseline by 2.4% and 1.8% in accuracy, respectively. These results

Table 4: In the 1-Shot comparison experiments conducted on the DomainNet dataset, the best results are highlighted in bold.

| Method | R→C | R→P | P→C | C→S | S→P | R→S | P→R | Avg. |
|---|---|---|---|---|---|---|---|---|
| S+T | 55.6 | 60.6 | 56.8 | 50.8 | 56.0 | 46.3 | 71.8 | 56.9 |
| DANNGanin et al. (2016) | 58.2 | 61.4 | 56.3 | 52.8 | 57.4 | 52.2 | 70.3 | 58.4 |
| ENTGrandvalet & Bengio (2004) | 65.2 | 65.9 | 65.4 | 54.6 | 59.7 | 52.1 | 75.0 | 62.6 |
| APEKim & Kim (2020) | 70.4 | 70.8 | 72.9 | 56.7 | 64.5 | 63.0 | 76.6 | 67.6 |
| DECOTAYang et al. (2021) | 79.1 | 74.9 | 76.9 | 65.1 | **72.0** | 69.7 | 79.6 | 73.9 |
| MMESaito et al. (2019a) | 70.0 | 67.7 | 69.0 | 56.3 | 64.8 | 61.0 | 76.1 | 66.4 |
| MME SLAYu & Lin (2023) | 71.8 | 68.2 | 70.4 | 59.3 | 64.9 | 61.8 | 77.2 | 68.8 |
| CDACLi et al. (2021) | 77.4 | 74.2 | 75.5 | 67.6 | 71.0 | 69.2 | 80.4 | 73.6 |
| CDAC SLAYu & Lin (2023) | 79.2 | 75.2 | **77.2** | 68.1 | 71.7 | 71.7 | 80.4 | 74.8 |
| AGLP(Ours) | **80.1** | **75.7** | **77.2** | **68.9** | 71.9 | **72.0** | **81.0** | **75.3** |

affirm the effectiveness of our approach in semi-supervised domain adaptation tasks across varying data availability scenarios.

**DomainNet:**To further validate the performance of our model, we conducted 1-Shot and 3-Shot experiments on the larger and more complex DomainNet dataset, with results summarized in Table. 3 and Table. 4. Our model achieved accuracies of 75.3% and 77.3%, outperforming all comparative methods. Specifically, compared to the baseline (CDAC SLA), the model's accuracy improved by 0.5% in the 1-Shot experiment and by 0.8% in the 3-Shot experiment. It is noteworthy that due to the larger and more complex nature of the DomainNet dataset, the performance improvements were less pronounced compared to those observed in Office-Home. Nevertheless, these results demonstrate that our model maintains strong performance even on more challenging datasets.

Table 5: Ablation experiments were conducted on the Office-Home 3-Shot experiment, with the best results indicated in bold.

| Method | Method *baseline* | SAA | CA | A→C | C→P | P→R | R→A | Avg. |
|---|---|---|---|---|---|---|---|---|
|  | ✔ | ✘ | ✘ | 65.6 | 82.1 | 82.6 | 69.0 | 74.8 |
|  | ✔ | ✔ | ✘ | 68.7 | **82.2** | 86.5 | 70.2 | 76.9 |
|  | ✔ | ✘ | ✔ | 67.4 | 81.7 | 85.4 | 69.8 | 76.1 |
| AGLP(Ours) | ✔ | ✔ | ✔ | **68.9** | 82.1 | **88.2** | **71.3** | **77.6** |

## 3.4 FURTHER PERFORMANCE ANALYSIS

### 3.4.1 ABLATION STUDY

To further validate the effectiveness of our model, we conducted an ablation study on Office-Home 3-Shot, as shown in Table 5. In this study, SAA refers to structure-aware alignment , and CA denotes class centroid alignment. As presented in Table 5, each component provides significant improvements over the baseline (CDAC SLA), although the enhancement from CA is less pronounced. This may be attributed to CA primarily optimizing the scores of structure-aware alignment. When both components are utilized together, optimal performance is achieved. Overall, our improvements are evidently effective and can be transferred to other models.

### 3.4.2 VISUALIZATION ANALYSIS

To more intuitively validate our model, we conducted various analyses during the Office-Home 3-Shot A→C domain adaptation experiment, including t-SNE dimensionality reduction visualization, confusion matrix evaluation, loss convergence, and accuracy comparison.

**Confusion Matrix:** The confusion matrix comparison in Figure 3 (a) and (b) highlights the performance of our model against the baseline (CDAC SLA). We calculated the confusion matrix for

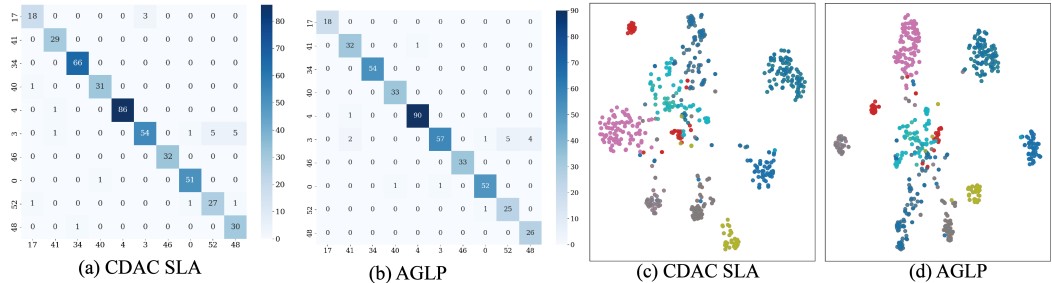

Figure 3: In the Office-Home 3-Shot A→C domain adaptation experiment, the confusion matrix and visualization analysis were computed by randomly selecting 10 classes from the dataset.

the 10 selected categories from the test samples, showing that our model achieves higher accuracy compared to the baseline.

**Dimensionality Reduction Visualization:** As shown in Figure 3 (c) and (d), we compared MME SLAYu & Lin (2023), CDACLi et al. (2021), CDAC SLAYu & Lin (2023), and our model. We randomly selected 10 categories from the 65 categories in Office-Home and extracted features using the trained model, subsequently reducing them to a two-dimensional space using t-SNE. Our model exhibits better clustering of sample features, demonstrating improved domain adaptation performance.

**Loss Convergence:** The loss convergence results are depicted in Figure 4 (a). Here, CA loss represents our improvement $\mathcal{L}_{CA}$, Source loss denotes $\tilde{\mathcal{L}}_s(g|S)$, Target loss corresponds to $\mathcal{L}_{CE}$, and Unlabeled loss represents $\mathcal{L}_u$. Our model demonstrates rapid convergence during training. Notably, $\tilde{\mathcal{L}}_s(g|S)$ experiences a spike due to warmup but subsequently converges effectively.

**Test Accuracy Comparison:** The accuracy variation results, shown in Figure 4 (b), compare our model with MME SLAYu & Lin (2023), CDAC, and CDAC SLA. Our model consistently maintains superior accuracy, confirming its excellent performance.

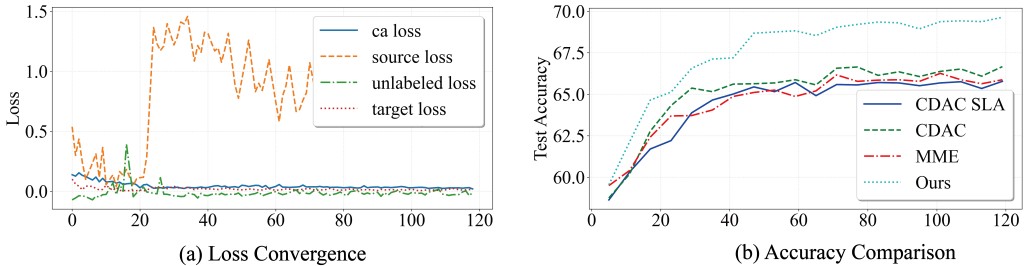

Figure 4: (a) illustrates the convergence behavior of the four loss functions in our model during the Office-Home 3-Shot A→C domain adaptation experiment. (b) depicts the accuracy variations of the four models throughout the same experiment.

## 4 CONCLUSION

In this paper, we propose a novel method by leveraging graph structure information in a unified network for semi-supervised domain adaptation. Our model introduces a class alignment loss to achieve this goal and employs a moving centroid strategy to mitigate the influence of incorrect pseudo-labels. To match source and target domain distribution robustly, we design an effective structure data alignment mechanism for SSDA. By modeling this alignment mechanism, the deep network can generate domain-invariant and highly discriminative semantic representations. Experiments on standard domain adaptation datasets verify the effectiveness of the proposed model.

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

# A APPENDIX

## A.1 RELATED WORK

### A.1.1 UNSUPERVISED DOMAIN ADAPTATION

Unsupervised Domain Adaptation (UDA) Ganin & Lempitsky (2015); Wang & Deng (2018); Liu et al. (2022) aims to adapt models trained on source domain data to unlabeled target domain data. Most domain adaptation algorithms attempt to align feature distributions by minimizing the domain shift between the source and target domains, facilitating the transfer of knowledge learned from source data to improve classification performance in the target domain Ghosn & Bengio (2003); Weiss et al. (2016). Many UDA methods employ a domain classifier to measure the distance Long et al. (2018); Shu et al. (2018). The domain classifier is trained to distinguish whether the input features originate from the source or the target, while the feature extractor is trained to deceive the domain classifier by matching feature distributions. Recently, some studies have addressed this issue by constructing an end-to-end mapping from the source domain to the target domain using clustering-based Optimal Transport Liu et al. (2023), and Yue *et al.* Yue et al. (2023) proposed Invariant Consistency Learning to tackle the spurious correlation between domain-specific features and class features.

UDA has now been applied in various domains, such as image classification Liu et al. (2022), semantic segmentation Sankaranarayanan et al. (2018), and object detection Saito et al. (2019b). It has also spawned derivative directions, utilizing multiple source domains to generalize to unseen target domains through domain generalization Wang et al. (2022); Zhou et al. (2022). Additionally, test-time training/test-time adaptation employs unlabeled target domain data only during the testing phase, without using source domain data Sun et al. (2020); Liu et al. (2021); Wang et al. (2020), and semi-supervised domain adaptation leverages a small amount of labeled target domain data along with a large amount of unlabeled data for transfer Saito et al. (2019a).

### A.1.2 SEMI-SUPERVISED DOMAIN ADAPTATION

Semi-Supervised Domain Adaptation (SSDA) aims to leverage a small number of labeled samples from the target domain, combined with source domain data and a large amount of unlabeled target domain data, significantly improving domain adaptation performance compared to Unsupervised

Domain Adaptation Saito et al. (2019a). Recently, SSDA has attracted widespread attention from researchers Kim & Kim (2020); Li et al. (2021); Yu & Lin (2023); Li et al. (2024), with relevant studies applying it to object detection to enhance performance Wang et al. (2023).

Saito et al. (2019a) addressed the SSDA problem by aligning features from both domains using adversarial learning. Yu & Lin (2023) proposed a novel source adaptation paradigm that treats the source domain as noisy target domain data, enhancing performance by cleaning label noise. Rahman et al. (2023) introduced a new semi-supervised domain adaptation framework utilizing autoencoders and synchronized learning to improve performance. Most prior methods have focused on sample-level feature alignment to tackle the SSDA problem. In this work, we aim to utilize Graph Convolutional Networks (GCNs) to capture structural information for aligning features from the source domain to the target domain.

### A.1.3 Graph on Domain Adaptation

Most domain adaptation frameworks are typically limited by their structure, often utilizing only domain labels and class labels, while neglecting important structural information from the data Ma et al. (2019). Ma et al. (2019) were the first to enhance the performance of Unsupervised Domain Adaptation through three alignment strategies: structure-aware alignment, domain alignment, and class centroid alignment. Zhu et al. (2021) introduced a novel graph for Unsupervised Adversarial Domain Adaptation (DA) that integrates sample-level and class-level structural information from both domains to improve performance. Ding et al. (2018) constructed a Graph Adaptive Knowledge Transfer (GAKT) model to jointly optimize target labels and domain-invariant features within a unified framework, thereby enhancing the performance of Unsupervised Domain Adaptation. Furthermore, Dai et al. (2022) proposed a novel graph transfer learning framework, AdaGCN, which leverages adversarial domain adaptation and graph convolutional techniques to enhance class-discriminative node representations and mitigate the differences between the source and target domains.

Overall, however, all existing research on graph structural information in domain adaptation has primarily focused on Unsupervised Domain Adaptation, with little application in Semi-Supervised Domain Adaptation.

### A.2 Parameter Robustness Analysis

Our approach is based on an extension of the CDAC SLA, where we selected the optimal parameters as reported in the original paper. To verify the robustness of our model regarding parameter sensitivity, we conducted a series of parametric experiments. These experiments were performed on the OfficeHome dataset, specifically on the 3-shot and 1-shot A→C tasks. We evaluated the impact of various parameters, including GCN's output channels (Table. 5), GCN layers (Table. 6), and $\beta$ (Table. 7), on the model's accuracy. During these experiments, other parameters were fixed at their optimal values to better isolate the effects of the parameters under investigation. The experimental results are shown in Table 5, 6 and 7 can be seen. As clearly illustrated in the figure, the model maintains good accuracy within a certain range of parameter values, confirming the robustness of our model across multiple parameters within the defined intervals.

### A.3 Complete Confusion Matrix Result

The complete confusion matrix results are presented in Figure 8. In addition to the four models mentioned in the main text, we also included experiments with S+T and ENT.

### A.3.1 Complete Dimensionality Reduction Visualization Result

The complete dimensionality reduction visualization results are presented in Figure 9. In addition to the four models mentioned in the main text, we also included experiments with S+T and ENT.

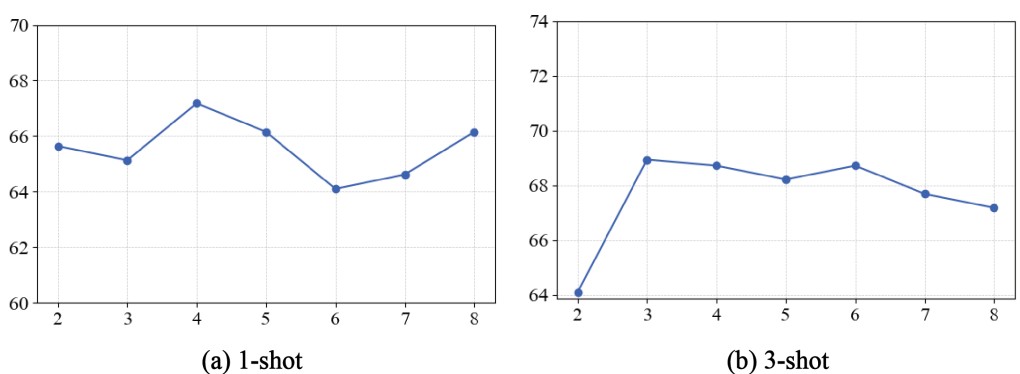

(a) 1-shot                    (b) 3-shot

Figure 5: In the domain adaptation experiment of Office-Home 3-Shot A→C, we conducted a parameter analysis by varying the output channels.

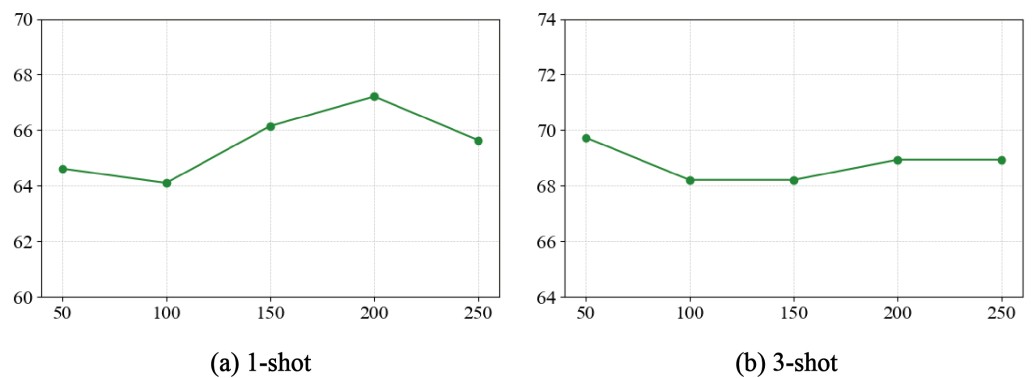

(a) 1-shot                    (b) 3-shot

Figure 6: In the domain adaptation experiment of Office-Home 3-Shot A→C, we conducted a parameter analysis by varying the GCN layers.

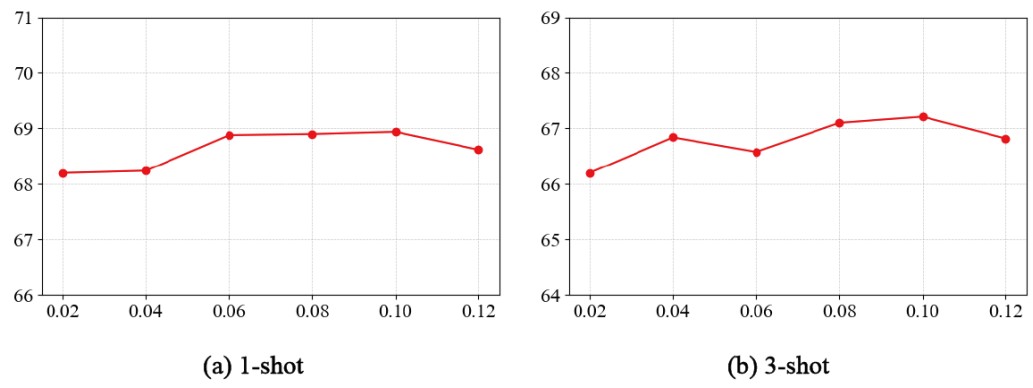

(a) 1-shot                    (b) 3-shot

Figure 7: In the domain adaptation experiment of Office-Home 3-Shot A→C, we conducted a parameter analysis by varying the $\beta$.

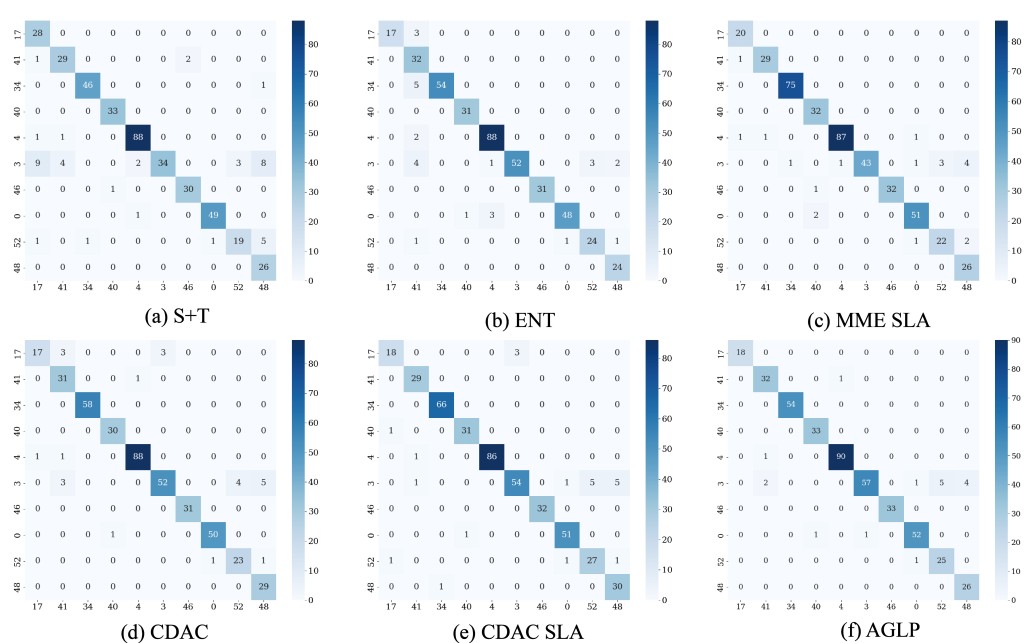

Figure 8: In the Office-Home 3-Shot A→C domain adaptation experiment, a confusion matrix was computed by randomly selecting 10 classes from the dataset.

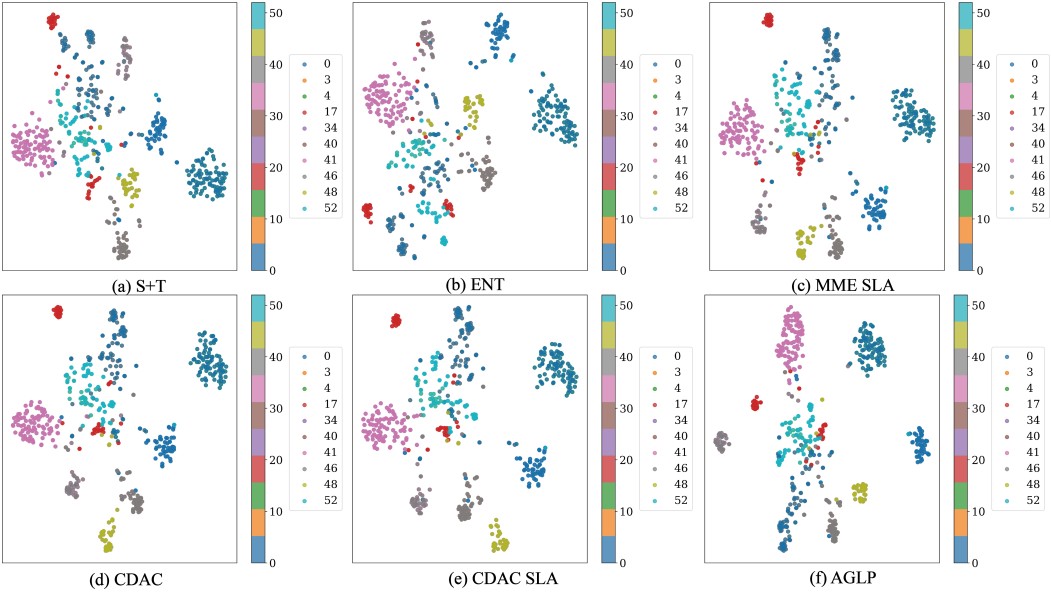

Figure 9: A visualization analysis was performed on the OfficeHome 3-Shot A→C domain adaptation experiment, where features were randomly extracted from 10 classes and reduced in dimensionality using t-SNE.

