# OpenReview forum: "AGLP: A Graph Learning Perspective for Semi-supervised Domain Adaptation"
_ICLR.cc/2025/Conference — ICLR 2025 Conference Withdrawn Submission_

### Official Review · Reviewer_xrZU · 2024-11-01

**Soundness:** 2
**Presentation:** 2
**Contribution:** 1
**Rating:** 3
**Confidence:** 4

**Summary:**

This paper proposes a graph learning perspective for the task of semi-supervised domain adaptation. Specifically, the structure-aware alignment method is proposed to capture the structural relationships between mini-batch source and target samples, as well as a class centroid alignment method is proposed to align the features of the same class. The experiments are conducted on the Office-Home and DomainNet datasets to demonstrate the effectiveness of the proposed methods.

**Strengths:**

1. The manuscript is well-orgnised. It is easy to follow this work.
2. The ablation studies of each module and hyperparameters are sufficient.

**Weaknesses:**

1. The novelty of this paper is limited, it seems to be a combination of existing methods. The structure-aware alignment method and class centroid alignment mentioned in the paper are similar to the existing work GCAN [1]. The method of learning data structure information using a graph convolutional network in the structure-aware alignment module, as well as the centroid alignment objective function in the class centroid alignment module are the same as in GCAN [1].
2. Most of the theories and equations in the paper are the same as those in other papers.
3. The Data Structure Analyzer module lacks a detailed explanation. How is the structure score obtained? It is suggested to provide a more detailed description of the Data Structure Analyzer module, including the specific algorithm or process used to calculate the structure scores.
4. The experimental results should be compared with more latest works, such as [2-3]. Are there any technical challenges in comparing to these methods?
5. This paper claims to be the first work to model graph information for semi-supervised domain adaptation. However, there have already been some graph-based semi-supervised domain adaptation works, such as [4-5]. It is suggested to clarify the difference between these existing graph-based SSDA methods.


[1] Ma, Xinhong, Tianzhu Zhang, and Changsheng Xu. "Gcan: Graph convolutional adversarial network for unsupervised domain adaptation." CVPR. 2019.
[2] He, Jiujun, Bin Liu, and Guosheng Yin. "Enhancing Semi-supervised Domain Adaptation via Effective Target Labeling." AAAI. 2024.
[3] Basak, Hritam, and Zhaozheng Yin. "Forget More to Learn More: Domain-Specific Feature Unlearning for Semi-supervised and Unsupervised Domain Adaptation.” ECCV.
[4] Li, Jinfeng, et al. "Domain-invariant graph for adaptive semi-supervised domain adaptation." ACM Transactions on Multimedia Computing, Communications, and Applications.2022.
[5] He, Gewen, et al. "Classification-aware semi-supervised domain adaptation." Proceedings of the IEEE/CVF conference on computer vision and pattern recognition workshops. 2020.

**Questions:**

How does the structure-aware alignment module align the structure-aware features? What is the objective function of this module that constrains the learning process?
How does the data structure analyzer in the structure-aware alignment module calculate the structure score?
What are the differences between the proposed proposed structure-aware alignment module and class centroid alignment module and the prior work GCAN[1]?

---

### Official Review · Reviewer_6r4q · 2024-11-01

**Soundness:** 3
**Presentation:** 3
**Contribution:** 3
**Rating:** 6
**Confidence:** 4

**Summary:**

The paper presents AGLP, an innovative graph learning approach designed for semi-supervised domain adaptation (SSDA). By integrating structural information from the data through an instance graph and applying a Graph Convolutional Network (GCN), AGLP seeks to align source and target domains to enhance generalization. The model also incorporates class centroid alignment and structure-aware alignment to achieve domain-invariant and semantically discriminative representations. Empirical results on the Office-Home and DomainNet benchmarks show that AGLP outperforms existing SSDA methods, demonstrating its effectiveness and robustness.

**Strengths:**

1.AGLP introduces a graph-based perspective in SSDA, effectively modeling structural information to address gaps in current SSDA techniques that primarily rely on domain and class labels without structural insights.
2. he model consistently achieves high accuracy across benchmarks, with ablation and comparative analyses that underline the impact of structure-aware and class centroid alignment mechanisms on performance.
3. This work addresses a vital aspect of SSDA—structural alignment—and introduces techniques that could influence further research in graph-based domain adaptation.
4. The methodology is described comprehensively, detailing the GCN construction, alignment losses, and their respective roles in achieving effective alignment and consistency.

**Weaknesses:**

1. The integration of GCNs and alignment mechanisms increases model complexity and computational overhead, which might present challenges in low-resource environments. Future work could explore ways to optimize computational efficiency.
2. While the graph structure construction is a strength of the approach, further clarification on hyperparameter tuning would enhance understanding, especially concerning batch size variability and data structure adaptation.

**Questions:**

1.Could the authors elaborate on the computational requirements for AGLP, particularly the demands associated with GCNs and alignment mechanisms?
2.How does AGLP manage inconsistencies that may arise from pseudo-labels, especially when the source and target domain structures vary significantly?

---

### Official Review · Reviewer_DZeL · 2024-11-01

**Soundness:** 2
**Presentation:** 2
**Contribution:** 2
**Rating:** 3
**Confidence:** 4

**Summary:**

The manuscript presents an end-to-end Graph Convolutional Adversarial Network (GCAN) for semi-supervised domain adaptation in classification tasks. It applies a Graph Convolutional Network (GCN) to an instance graph, allowing structural information to propagate along weighted graph edges that can be learned within the designed network, enabling the network to better capture structural information in features. The method’s effectiveness is validated on multiple standard benchmarks.

**Strengths:**

The manuscript introduces a graph learning perspective to semi-supervised domain adaptation (SSDA) and leverages structural information in instance graphs, which is a relatively novel approach. By using class centroid alignment to reduce inter-class discrepancies, the overall design is quite reasonable. The method demonstrates performance improvements compared to other approaches, which, to some extent, validates its effectiveness.

**Weaknesses:**

1.	The manuscript proposes using a graph convolutional network (GCN) for structural information propagation; however, it does not clearly explain why this form of structural propagation effectively reduces domain discrepancies. The overall description lacks clarity, and the innovation points are not accurately conveyed.
2.	The manuscript proposes constructing a densely connected instance graph using CNN features of samples, connecting them based on the similarity of their structural characteristics. However, it does not provide a clear rationale for how this instance graph construction reduces domain discrepancies. Additionally, the reasoning behind the use of class centroid alignment to enable the learned representations to effectively encode category label information is not well articulated. A more in-depth explanation of these mechanisms would strengthen the manuscript by clarifying how these design choices directly contribute to reducing domain gaps and improving representation learning for SSDA tasks.
3.	In the experimental section, structural information or feature visualizations should be added to verify the effectiveness of the method.

**Questions:**

The manuscript needs to further elaborate on how GCN can effectively reduce inter-domain differences and discuss in more detail how GCN enhances the extraction of domain-invariant features, particularly its underlying mechanism in semi-supervised domain adaptation.

---

### Official Review · Reviewer_uDTs · 2024-11-02

**Soundness:** 2
**Presentation:** 2
**Contribution:** 2
**Rating:** 3
**Confidence:** 4

**Summary:**

The paper proposes a graph learning perspective (AGLP) for semi-supervised domain adaptation (SSDA). The proposed AGLP model leverages the graph convolutional network to capture the structural information of the data and learn domain-invariant and semantic representations. The experimental results on multiple benchmarks demonstrate that AGLP outperforms state-of-the-art SSDA methods.

**Strengths:**

(1) The paper introduces a graph learning perspective for SSDA, which is the first work to model structural information in SSDA.
(2) The proposed AGLP model effectively learns domain-invariant and semantic representations, reducing domain discrepancies in SSDA.
(3) The experimental results show that AGLP outperforms state-of-the-art SSDA methods on multiple benchmarks.

**Weaknesses:**

(1) The motivation of this manuscript is not clear. The authors should clearly claim the challenging issues in previous methods.
(2) While the GNN adopted in the manuscript seems plausible, it is not exciting. Although the authors emphasize that this is the first semi-supervised domain adaptation method using graph networks, graph networks have been widely used in more difficult unsupervised domain adaptation tasks.
(3) While the graph module adopted in the manuscript seems plausible, it is not exciting.

**Questions:**

See Weaknesses

---

### Note · Authors · 2024-11-12

I have read and agree with the venue's withdrawal policy on behalf of myself and my co-authors.